# Strong Opponent of Walnut Anthracnose—*Bacillus velezensis* and Its Transcriptome Analysis

**DOI:** 10.3390/microorganisms11081885

**Published:** 2023-07-26

**Authors:** Linmin Wang, Tianhui Zhu

**Affiliations:** College of Forestry, Sichuan Agricultural University, Yaan 625000, China

**Keywords:** *Bacillus velezensis*, colletotrichum gloeosporioides, transcriptional regulation

## Abstract

Walnut is a significant economic tree species worldwide. Walnut anthracnose, caused by the pathogen *Colletotrichum gloeosporioides*, greatly reduces walnut production and economic benefits. Our study showed that *Bacillus velezensis* effectively halted the growth of *C. gloeosporioides*, inducing noticeable abnormalities such as hyphal breakage and distortion, thereby curtailing the pathogen’s virulence. A 50–100 times dilution of *B. velezensis* fermentation broth, applied every two to three days, served as an efficient protective layer for walnut leaves and fruits against *C. gloeosporioides* infection. Transcriptomic analysis of *B. velezensis* unveiled its dynamic response against *C. gloeosporioides*. On the second day, *B. velezensis* upregulated a significant number of differentially expressed genes related to the synthesis of metabolic products, amino acid biosynthesis, and motility. On the fourth day, continuous synthesis of metabolic products and amino acids, along with differential expression of spore-related genes, was observed. By the sixth day, the focus shifted towards environmental adaptation and carbon source utilization. Throughout the process, *B. velezensis* likely employed strategies such as the release of metabolic products, increased chemotaxis, and nutrient competition to exert its antagonistic effect on *C. gloeosporioides*. Fluorescence quantitative results showed that 15 primer pairs were up-regulated and 15 were down-regulated, with a 100% similarity rate to transcriptome sequencing results, confirming their authenticity. These findings provided a foundation for the widespread application of *B. velezensis* as a biocontrol agent in agriculture and forestry.

## 1. Introduction

Walnut, a deciduous tree species, holds significant economic value, with extensive applications in food, pharmaceuticals, cosmetics, and other industries [1,2]. Major walnut-producing countries across the globe include the United States, Iran, Turkey, China, Mexico, Ukraine, France, Italy, Spain, and Chile [3,4,5]. China, in particular, stands as one of the world’s largest walnut cultivators, with a rich history of walnut farming that dates back thousands of years [6]. However, the walnut industry faces significant economic losses due to various diseases, particularly anthracnose, which is caused by the pathogen *Colletotrichum gloeosporioides* (*C. gloeosporioides*). This disease, characterized by its rapid onset and swift transmission, can lead to fruit rot, leaf wilt, branch death, and in severe cases, plant death [7,8]. The current primary management strategy is chemical control [9], yet this approach has several drawbacks, including environmental pollution [10], pathogen resistance [11,12], and risks to non-target organisms [13,14]. Consequently, the exploration of environmentally friendly and efficient solutions such as *Bacillus velezensis* (*B. velezensis*) is crucial. *B. velezensis*, a member of the *Bacillus* genus, is renowned for its broad-spectrum antimicrobial properties, ability to produce bioactive substances, and adaptability to various environments [15,16,17,18]. For instance, *B. velezensis* 1B-23 and 1D-12 combat bacterial canker in tomatoes by producing surface proteins A, B, and C, while *B. velezensis* Bv-21 prevents citrus ulcer disease with an ethyl acetate extract [19,20]. Another strain, CC09, mitigates wheat diseases caused by dwarf bacteria and sorghum double spore via the secretion of antagonistic metabolic products [21]. The applications of *B. velezensis* extend beyond pathogen control. As reported by Nanjani et al., *B. velezensis* K1 not only combats the fungal pathogen *Alternaria brunsii* but also serves as a biofertilizer, aiding plants deficient in iron and phosphate [22]. Additionally, Wei et al. demonstrated that *B. velezensis* YW17 could control ginseng root rot disease by both producing antifungal substances and regulating the root and rhizosphere microbial communities [23].

In recent years, transcriptomic studies on *Bacillus* have led to significant discoveries in biocontrol, offering new insights into its application for biological defense mechanisms [24,25,26]. *B. velezensis* has been found to induce plant immunity through specific signaling pathways and modulate the expression of key genes to enhance its survival and resistance against fungal pathogens. For instance, it has been shown to elicit plant immunity via the intricate interplay between the ethylene (ET), jasmonic acid (JA), and salicylic acid (SA) signaling pathways and the brassinosteroid biosynthesis pathway [27,28,29]. Moreover, *B. velezensis* strains, when co-cultivated with pathogenic fungi such as *Fusarium graminearum*, have been found to upregulate genes related to sporulation and phosphate stress, while downregulating those associated with secondary metabolism, biofilm formation, and the tricarboxylic acid cycle [30]. A study by Pandin et al. showed upregulation of certain genes, including surfactin (srfAA) and fengycin (fenA), when *B. velezensis* was confronted with fungal pathogens in compost [31]. Furthermore, *B. velezensis* has been demonstrated to enhance plant resistance to specific pests via unique chemical pathways. For instance, *B. velezensis* YC7010 can trigger systemic resistance in rice seedlings against *Nilaparvata lugens Stål* via SA and JA-dependent pathways [32]. Transcriptomics technology has further substantiated *B. velezensis*’s capability to produce metabolic byproducts that suppress the growth of specific pathogens. A study by Li et al. showed that the crude extract of *B. velezensis* fermentation broth significantly induced differential gene expression in pathogens, affecting various biological processes [33]. Moreover, Zhang et al.’s study clarified the mechanism by which lipopeptides produced by *B. velezensis* GS-1 inhibit the growth of *Magnaporthe oryzae* [34]. Despite significant advancements in the field of transcriptome analysis, a substantial knowledge gap remains, particularly in the study of bacterial transcriptomes. Current research predominantly focuses on fungal transcriptomes, leaving studies on bacteria, especially those involving interactions between *B. velezensis* and *C. gloeosporioides*, relatively underexplored [35]. In this study, we established an interaction system to examine the control exerted by *B. velezensis* on *C. gloeosporioides* at various time points (2, 4, and 6 days) on a petri dish and its impact on virulence. At the same time, we analyzed the transcriptome information and differentially expressed gene enrichment of *B. velezensis* at these time points. Additionally, we conducted leaf control experiments to study the control effects of *B. velezensis* on *C. gloeosporioides* at different concentrations and frequencies. The goal of this research is to identify key genes related to antagonism through transcriptome analysis, in order to further investigate the biocontrol mechanism of *B. velezensis* against *C. gloeosporioides*. Through this study, we evaluated the protective effect of *B. velezensis* on walnut leaves and fruits, providing beneficial strategies for walnut cultivation, and laying a foundation for the development and application of biological control agents.

## 2. Materials and Methods

### 2.1. Tested Fungal, Bacteria and PLANT Materials

*C. gloeosporioides* and *B. velezensis* was provided by Forest Protection Laboratory, Forestry College of Sichuan Agricultural University. *B. velezensis* was isolated from the soil under walnut trees at the Chongzhou base of Sichuan Agricultural University, Chengdu City, Sichuan Province (longitude and latitude: 103.65169, 30.56266, altitude: 534 m). The walnut leaves were obtained from three-year-old walnut trees planted in Chengdu Academy of Agricultural and Forestry Sciences, Sichuan Province (longitude and latitude: 104.07275, 30.57899, altitude: 534 m). The walnut leaves for the experiment were carefully selected from branches of the same age and position on the tree, ensuring almost identical size and shape. The leaves were healthy, without any disease spots. The walnut fruits were collected from the same location, Chongzhou base of Sichuan Agricultural University, and they were of similar size, with intact peels and no signs of pests or diseases.

### 2.2. Effect of B. velezensis on Pathogenicity of C. gloeosporioides

An interaction system between *B. velezensis* and *C. gloeosporioides* was established using the plate confrontation method. A 4.0 mm diameter fungal cake of *C. gloeosporioides* was inoculated at the center of the PDA medium, while the *B. velezensis* strain was inoculated at four equidistant points 25.00 mm from the center. A control plate with only *B. velezensis* was also prepared. Each set had three replicates, cultured at 28 °C, and each set was observed on days 2, 4, and 6. On day 6, the mycelium of *C. gloeosporioides* in both the treatment and control groups was examined and photographed using an Olympus BX43 microscope (Olympus BX43, Tokyo, Japan).

*C. gloeosporioides* mycelium, after interacting with *B. velezensis* for 2, 4, and 6 days, was punched from the edge and inoculated onto walnut leaves (D2_T, D4_T, and D6_T groups). Mycelium not confronting *B. velezensis* for 2, 4, and 6 days was used as a blank control (D2_W, D4_W, and D6_W groups). Three replicates were cultured at 25 °C. Prior to inoculation, the leaves were washed with sterile water and disinfected with 70% ethanol spray. On the 12th day post-inoculation, the lesion area of the walnut leaves and the relative inhibition rate of *B. velezensis* against *C. gloeosporioides* were measured and calculated.
$$I(\%)=\frac{\left(C\;-\;T\right)}{C}\times 100$$

I: inhibition rate; C: the leaf spot area of negative control (cm^2^); T: the leaf spot area of treatment (cm^2^).

### 2.3. Control Effect of B. velezensis against C. gloeosporioides on Detached Leaves and Fruits

*C. gloeosporioides* was cultured on PDA solid medium at 28 °C for 4 days. Healthy walnut leaves of the same age, position on the branch, and similar size were selected for biocontrol experiments. Prior to the inoculation trials, the walnut leaves were washed with sterile water and disinfected with 70% ethanol spray. Each walnut leaf was inoculated with 2 *C. gloeosporioides* cakes, which were obtained by punching the edge of the 4-day-old culture on PDA medium. The single colony of *B. velezensis* was inoculated in LB liquid medium and cultured at 28 °C, 180 rpm for 24 h. Subsequently, the *B. velezensis* fermentation broth was diluted with sterile water until an OD_600_ of 1.0 (106 Colony Forming Unit/mL) was reached. Starting from the high concentration, seven concentration gradients were set: 1, 10, 50, 100, 300, 600, and 1200 times. The treatment groups were named as T_1, T_10, T_50, T_100, T_300, T_600, and T_1200. The inoculated walnut leaves were sprayed once every 48 h, and the control group was sprayed with the same amount of LB liquid medium, with 3 replicates in each group. The detailed processing methods are shown in Appendix A (provided in Appendix A). After 14 days of continuous treatment, the control effect of *B. velezensis* on walnut anthracnose was observed and recorded on the 15th day.

After determining the optimal control concentration, the experiment was conducted using 5 spray frequencies: every 1 day (namely T1 group), 2 days (namely T2 group), 3 days (namely T3 group), 4 days (namely T4 group), and 5 days (namely T5 group). The CK group was sprayed daily with LB liquid medium as a control. The leaves were sprayed at the same time each day, using the best spray concentration (50 times). In between treatment sprays, all groups received LB liquid medium. The detailed processing methods are shown in Appendix A (provided in Appendix A). Each group had 3 replicates, and the experiment lasted for 15 days, with leaf lesion area changes observed and recorded.

*C. gloeosporioides* was cultured on PDA solid medium at 28 °C for 4 days, and the edge of the mycelium was punched out. The punched fungus cakes were inoculated onto walnut fruit surfaces. For the experiment, walnut fruits were sourced from the same walnut tree, ensuring that they were nearly identical in size and free of any disease or lesions. Treatment and control groups were established, with the treatment group receiving the optimal control concentration (50 times) and frequency (every 2 days) of fermentation broth, while the control group was sprayed with LB liquid medium at the same frequency. Each group had three replicates. Over an 18-day observation period, the changes in walnut lesions were monitored and recorded.

### 2.4. Transcriptional Analysis of B. velezensis

In method 2.2, the interaction system between *B. velezensis* and *C. gloeosporioides* was established, with three replicates per group. *B. velezensis* strains interacting with *C. gloeosporioides* for 2, 4, and 6 days were collected (namely D2_D, D4_D, and D6_D groups), while non-interacting strains served as controls (namely D2_CK, D4_CK, and D6_CK groups). Under aseptic conditions, a sterile inoculating loop was used to transfer *B. velezensis* colonies into nuclease-free centrifuge tubes containing PBS buffer. The samples were stored at −80 °C for transcriptome sequencing.

#### 2.4.1. RNA Extraction, Library Preparation and Sequencing

RNA was isolated using the TianGen DP430 Bacterial RNA Kit (TianGen, BeiJing, China) according to the manufacturer’s instructions. RNA degradation and contamination was monitored on 1% agarose gels. RNA integrity was assessed using the RNA Nano 6000 Assay Kit of the Bioanalyzer 2100 system (Agilent Technologies, Santa Clara, CA, USA). Total RNA was used as input material for the RNA sample preparations. For prokaryotic samples, mRNA was purified from total RNA using probes to remove rRNA. Fragmentation was carried out using divalent cations under elevated temperature in First Strand Synthesis Reaction Buffer (5×). First strand cDNA was synthesized using random hexamer primer and M-MuLV Reverse Transcriptase, then RNaseH was used to degrade the RNA. In the DNA polymerase I system, dUTP was used to replace the dNTP of dTTP as the raw material to synthesize the second strand of cDNA. The remaining overhangs were converted into blunt ends via exonuclease/polymerase activities. After adenylation of 3′ ends of DNA fragments, the adaptors with hairpin loop structure were ligated to prepare for hybridization. Then, USER Enzyme was used to degrade the second strand of cDNA containing U. In order to select cDNA fragments preferentially of 370~420 bp in length, the library fragments were purified with AMPure XP system (Beckman Coulter, Beverly, TX, USA). Then, PCR was performed with Phusion High-Fidelity DNA polymerase, Universal PCR primers and Index (X) Primer. Finally, PCR products were purified (AMPure XP system) and library quality was assessed on the Agilent Bioanalyzer 2100 system. The clustering of the index-coded samples was performed on a cBot Cluster Generation System using TruSeq PE Cluster Kit v3-cBot-HS (Illumia, San Diego, CA, USA) according to the manufacturer’s instructions. After cluster generation, the library preparations were sequenced on an Illumina Novaseq platform and 150 bp paired-end reads were generated.

#### 2.4.2. Quality Control and Data Analysis

Raw data (raw reads) of the fastq format were firstly processed through in-house perl scripts. In this step, clean data (clean reads) were obtained by removing reads containing adapter, reads containing N base, and low quality reads from raw data. At the same time, Q20, Q30, and GC content, the clean data were calculated. All the downstream analyses were based on the clean data with high quality. The reference genome and annotation files were downloaded from publicly available data submitted to NCBI (https://ftp.ncbi.nlm.nih.gov/genomes/all/GCF/002/117/165/GCF_002117165.1_ASM211716v1/, accessed on 2 May 2017. The assembly number was GCF_002117165.1.). Both building index of reference genome and aligning clean reads to reference genome were used Bowtie2-2.2.3.

#### 2.4.3. Quantitative and Differential Expression Analysis of Gene Expression

HTSeq v0.6.1 was used to count the reads numbers mapped to each gene. Then, the FPKM of each gene was calculated based on the length of the gene and the reads count mapped to the gene. Differential expression analysis of three groups (three biological replicates per condition) was performed using the DESeq R package (1.18.0). DESeq provide statistical routines for determining differential expression indigital gene expression data using a model based on the negative binomial distribution. The resulting *p*-values were adjusted using the Benjamini and Hochberg’s approach for controlling the false discovery rate. Genes with an adjusted *p*-value < 0.05 found by DESeq were assigned as differentially expressed. Gene Ontology (GO) enrichment analysis of differentially expressed genes (DEGs) was implemented by the GOseq R package, in which gene length bias was corrected. GO terms with corrected *p*-value less than 0.05 were considered significantly enriched by differential expressed genes. The Kyoto Encyclopedia of Genes and Genomes (KEGG) is a database resource for understanding high-level functions and utilities of the biological system, such as the cell, the organism, and the ecosystem, from molecular-level information, especially large-scale molecular datasets generated by genome sequencing and other high-throughput experimental technologies (http://www.genome.jp/kegg/, accessed on 5 March 2021). We used KOBAS 3.0 software to test the statistical enrichment of differential expression genes in KEGG pathways. The DEGs were subjected to protein–protein interaction (PPI) analysis using the STRING database, which provides known and predicted PPI information. The gene names were input into the STRING protein interaction analysis platform, and multiple proteins were selected. The species was specified as *B. amyloliquefaciens* to obtain the PPI network of protein interactions. The confidence level was set to 0.9, and data were obtained after excluding invalid proteins that did not interact with other proteins. The resulting data was imported into Cytoscape 3.6.0, and the network analyzer function was used to calculate the degree of interaction for each protein. Based on the three highest degree values, the core targets were identified, and node size was determined by the degree value. Finally, a PPI network diagram was constructed.

#### 2.4.4. Quantitative Real-Time PCR

Thirty differentially expressed transcripts identified by RNA-Seq analysis were selected for gene expression analysis using quantitative real-time PCR (qRT-PCR). The sequences of the primers used are listed in Appendix A (provided in Appendix A). Each treatment group (D2_DvsD2_CK, D4_DvsD4_CK, and D6_DvsD6_CK) included ten candidate genes. Glyceraldehyde-3-phosphate dehydrogenase (GAPDH), a widely recognized housekeeping gene, was selected as the reference gene. Its ubiquitous and stable expression in cells ensures a dependable baseline for comparing the expression levels of other target genes. The target gene expression was normalized to GAPDH [36]. The transcriptional levels of the different genes were investigated and compared using the 2^−ΔΔCt^ method [37].

## 3. Results

### 3.1. Effect of B. velezensis on Mycelial Growth and Pathogenicity of C. gloeosporioides

Figure 1 effectively demonstrates the inhibitory effect of *B. velezensis* treatment on *C. gloeosporioides* growth over 2, 4, and 6 days. The treated mycelia consistently appeared smaller and more constrained compared to the untreated mycelia, which exhibited more extensive and circular growth patterns. By day 6, the untreated mycelia occupied the entire PDA medium (Figure 1G), while the treated mycelia maintained a diamond shape and remained distant from *B. velezensis* colonies (Figure 1I). The morphological characteristics of *C. gloeosporioides* mycelium observed on day 6 (Figure 2) further emphasized the differences between the control and treated groups, showcasing distinct variations in hyphal morphology. The control group hyphae were straight and thick, whereas the treated group hyphae displayed elongated, swollen, and twisted characteristics, with the majority of mycelial branches exhibiting twisted growth patterns. At the same time, it could also be observed that the shape of *B. velezensis* in the treatment group was smaller than that in the control group (Figure 1). We hypothesized that this might have been because *B. velezensis* sacrificed a portion of its nutritional growth to produce more antimicrobial substances, in an effort to counteract the infection of *C. gloeosporioides*.

Edge hyphae of *C. gloeosporioides* from both the control and treatment groups were utilized to infect walnut leaves on days 2, 4, and 6. As shown in Figure 3, a consistent observation across the control groups (D2_W, D4_W, D6_W) was the complete envelopment of the fungus cakes by *C. gloeosporioides*. The hyphae of *C. gloeosporioides* in these groups appeared to be profuse and robust, resulting in heightened leaf yellowing and a more extensive disease spot areas on the walnut leaves. Conversely, in the treatment groups (D2_T, D4_T, and D6_T), the amount of *C. gloeosporioides* hyphae was noticeably less than in the control group, and the hyphae did not fully cover the fungus cakes. Moreover, the hyphae appeared clustered and less spread out, possibly due to morphological abnormalities that obstruct their growth and expansion following confrontation culture. Consequently, the ability of *C. gloeosporioides* to infect leaves was diminished, resulting in significantly smaller disease areas in the treatment groups (D2_T, D4_T, and D6_T). The data analysis from Figure 4A further substantiated these findings, indicating that at each time point, the control groups consistently had larger disease spot areas compared to the treatment groups (*p* < 0.05). This suggested that the infectivity of *C. gloeosporioides* was significantly reduced following treatment with *B. velezensis*. Our analysis indicated that the inhibitory effect of *B. velezensis* on *C. gloeosporioides* was most potent after six days of exposure, followed by two days (Figure 4B). Although the inhibition rate at four days was relatively the weakest, it was still statistically significant compared to the blank control (*p* < 0.05). These findings demonstrated that *B. velezensis* could significantly reduce the infectivity of *C. gloeosporioides*, thereby achieving the goal of protecting the host.

### 3.2. Control Effect of B. velezensis on Detached Leaves and Fruits of Walnut

The effects of spraying different concentrations of *B. velezensis* fermentation broth on inhibiting walnut anthracnose infection caused by *C. gloeosporioides* were examined. Seven concentration gradients were used: 1, 10, 50, 100, 300, 600, and 1200 times. The treatment groups were labeled as T_1, T_10, T_50, T_100, T_300, T_600, and T_1200, with CK serving as the control. The results indicate that infection with the *C.gloeosporioides* manifested as gradual fading of green and yellow coloration around the walnut leaves, forming nearly circular black spots by the 7th or 9th day. Over time, lesions expanded, and in some cases, eventually covered the entire leaf (Figure 5). The Levene homogeneity test confirmed that the data met the precondition of homogeneity of variance, permitting the use of repeated measures ANOVA to examine the significance of lesion area change trends. Since the data failed the sphericity test (*p* < 0.05), the multivariate test results were utilized as a basis. According to these results, the time point effect was significant (F (4,13) = 577.243, *p* < 0.05), indicating notable changes in lesion area over time. The interaction effect of time point and concentration was also significant (F (7,16) = 455.141, *p* < 0.05), implying that these factors influenced one another. As shown in Figure 6A, the lesion area in each group continued to expand over time, with the rate of expansion increasing as time went on. According to the results (Figure 6B), the T_50 and T_100 groups showed the most effective control, with significantly lower average lesion areas of 0.449 cm^2^ and 0.68 cm^2^, respectively (*p* < 0.05). In contrast, the control group had the highest average lesion area of 8.938 cm^2^. The T_1 group exhibited some control effect, reducing the lesion area by 54.576% compared to the CK group, but higher concentrations hindered optimal control. Interestingly, as the fermentation fluid concentration decreased from T_1 to T_10, T_50, and T_100, the control effectiveness improved. However, a turning point was observed at the T_300 group, where control effectiveness started to decline. Further reduction in fermentation fluid concentration resulted in increased lesion areas. The T_600 and T_1200 groups had the highest concentrations and showed the weakest control effect, but still achieved significant reduction in lesion areas compared to the CK group (*p* < 0.05), with reductions of 23.395% and 12.251%, respectively. In summary, the T_50 and T_100 groups demonstrated the best control, followed by T_10, T_300, and T_1. While the T_600 and T_1200 groups had the weakest control effect, some level of disease prevention was still observed.

After inoculation with *C. gloeosporioides*, different frequencies of *B. velezensis* spray were tested on walnut leaves, with five gradients: T1, T2, T3, T4, and T5, representing spraying every 1 day, 2 days, 3 days, 4 days, and 5 days, respectively. The CK group served as a blank control, and each treatment had three replicates. Figure 7 clearly demonstrated that spraying walnut leaves with the same concentration of *B. velezensis* fermentation broth at different frequencies significantly impacted the infection degree of *C. gloeosporioides* on walnut leaves and controlled the expansion rate of the walnut leaf lesion area. The most pronounced effect was observed in the CK group, which exhibited the fastest chlorosis yellowing and lesion expansion rate. In contrast, the T2 and T3 groups showed minimal yellowing and significantly smaller lesion expansion. Further measurement and statistical analysis of the lesion areas were performed using repeated measures ANOVA. The lesion area data passed the Levene homogeneity test, meeting the precondition of homogeneity of variance. Since the data did not pass the sphericity test (*p* < 0.05), multivariate test results were employed. According to the multivariable results, the time point effect of Roy’s largest root was significant (F (4,9) = 507.519, *p* < 0.05), indicating notable changes in leaf lesion area over time. The interaction effect of the time point and treatment concentration of Roy’s largest root was also significant (F (5,12) = 524.656, *p* < 0.05), suggesting an interaction effect between these factors. As shown in Figure 8A, the damage area in the CK group began to expand after 7 days, while the damage areas in the T5, T1, and T4 treatment groups significantly increased after 11 days. This indicated that *B. velezensis* treatment could delay the expansion speed and onset time of walnut leaf lesions. According to the repeated analysis method estimating boundary value results (Figure 8B), the T2 and T3 groups demonstrated the most effective control, showing the least yellowing and significantly smaller lesion areas of only 0.737 cm^2^ and 1.008 cm^2^, respectively. In contrast, the T5 group had the weakest control effect. However, compared to the CK group, the T5 group still exhibited a significant control effect, with a reduction in lesion area by 21.867% (*p* < 0.05). Overall, the T1, T2, T3, T4, and T5 groups all effectively delayed lesion expansion and mitigated further infection of walnut leaves by *C. gloeosporioides*.

An in vitro control effect experiment was conducted on healthy, disease-free walnut fruits. As illustrated in Figure 9, *C. gloeosporioides* strongly infects walnuts in the control group. Over time, the walnut surface formed dark brown spots around the fungus cake, and the epidermis became rotten and depressed. Conversely, no obvious lesion formation occurred on the walnut fruit surface after spraying with *B. velezensis*. The results indicate that *B. velezensis* fermentation broth can effectively prevent *C. gloeosporioides* infection on walnut fruit.

### 3.3. Transcriptome Results

#### 3.3.1. Sequencing Data Statistics

The transcriptome data of the *B. velezensis* has been reserved in the NCBI’s Sequence Read Archive (SRA) database under the accession number PRJNA742867. All experiments were implemented in triplicate, and the experimental data were expressed as the means ± SD. In total, 138,383,924 clean reads were obtained, and the percentage of Q30 bases in each sample was more than 93.28%. The percentage of GC content was in a range of 46.98–54.86%. The transcriptome sequencing results met the quality requirements of subsequent assembly analysis and the samples had high quality and high expression. The sequencing data of samples are shown in Appendix A (provided in Appendix A).

#### 3.3.2. Analysis of Gene Expression

The differential gene expression analysis was conducted among the D2_D, D4_D, and D6_D groups compared to their respective control groups (D2_CK, D4_CK, and D6_CK). Figure 10 illustrates the differences in expression levels between the samples. The numbers of DEGs were relatively similar between the D4_DvsD4_CK and D6_DvsD6_CK groups. Among the three groups, the D4_DvsD4_CK group exhibited the fewest up-regulated genes, while the D6_DvsD6_CK group had the most up-regulated genes. Conversely, the D2_DvsD2_CK group had the largest number of DEGs when compared to the other two groups. This finding suggests that *B. velezensis* displayed antibacterial activity against *C. gloeosporioides* from day 2 onwards, despite the considerable distance between the two strains on the substrate.

#### 3.3.3. Gene Annotation Results

The unigene sequences of *B. velezensis* obtained under different treatment conditions were analyzed using GO and KEGG databases (Figure 11). The GO analysis classified the unigenes into biological processes (BP), cellular components (CC), and molecular functions (MF), with DEGs predominantly enriched in BP and MF. Distinct patterns of enrichment were observed across the comparison groups D2_DvsD2_CK, D4_DvsD4_CK, and D6_DvsD6_CK, indicating *B. velezensis* activates various genes at different stages to cope with *C. gloeosporioides* infection. In the D2_DvsD2_CK group, DEGs were significantly enriched in locomotion, non-membrane-bounded organelle, and organelle functions. This suggests that *B. velezensis* may modify its cellular structure and motility at this stage to counteract the attack of *C. gloeosporioides*. Particularly in the D2_DvsD2_CK group, the highest number of enriched DEGs were observed, indicating that *B. velezensis* stimulates numerous genes during this phase to respond to *C. gloeosporioides* infection. The D4_DvsD4_CK group showed DEGs primarily enriched in CC and MF, with significant enrichment in the aromatic amino acid family metabolic process. This indicates that *B. velezensis* might enhance the synthesis of aromatic amino acids at this stage, which play a crucial role in producing antimicrobial compounds and inhibiting the growth of *C. gloeosporioides*. In the D6_DvsD6_CK group, DEGs were predominantly enriched in BP and MF, with significant enrichment in pathways related to carbohydrate transport and metabolism. This suggests that *B. velezensis* may adjust these processes at this stage to resist *C. gloeosporioides* infection, while phosphorylated proteins may play a key role in regulating these processes.

The KEGG functional enrichment analysis identified the top 20 most significantly enriched pathways for each comparison group. Amino acid biosynthesis pathways were consistently enriched in both the D2_DvsD2_CK and D4_DvsD4_CK groups, suggesting that *B. velezensis* may enhance its production of essential amino acids under these conditions, potentially contributing to the synthesis of antimicrobial peptides or other secondary metabolites with antifungal properties. Energy metabolism pathways, including TCA cycle, and glycolysis/gluconeogenesis, were significantly enriched in both the D2_DvsD2_CK and D6_DvsD6_CK groups. This observation implies that *B. velezensis* may increase its energy production and metabolic activity under these conditions, enabling a more robust response against *C. gloeosporioides* infection. Phenylalanine, tyrosine, and tryptophan biosynthesis, and propanoate metabolism pathways were enriched in both the D4_DvsD4_CK and D6_DvsD6_CK groups. These pathways are involved in the synthesis of aromatic amino acids and organic acids, which may play a role in the production of antimicrobial compounds or altering the local environment to inhibit the growth of *C. gloeosporioides*.

#### 3.3.4. PPI Network Analysis and qRT-PCR Verification

The PPI network analysis of DEGs in *B. velezensis* across various stages and groups revealed key gene interactions involved in biocontrol mechanisms (Figure 12). In the early stage (D2_DvsD2_CK group), the PPI network consisted of 159 nodes and 1276 edges. Core genes such as rpmD, rpsG, and rplB were significantly enriched in pathways such as ribosome, biosynthesis of secondary metabolites, biosynthesis of amino acids, biosynthesis of cofactors, flagellar assembly, carbon metabolism, microbial metabolism in diverse environments, and biotin metabolism. Notably, ribosome, biosynthesis of secondary metabolites, and flagellar assembly showed the highest gene enrichments. This suggests that *B. velezensis* likely produces antimicrobial secondary metabolites, enhances flagellar assembly and motility, and boosts protein synthesis capacity. In the mid-stage (D4_DvsD4_CK group), the PPI network had 54 nodes and 112 edges. The core genes, including acoL, hisH, and ilvC, among others, were significantly enriched in pathways such as biosynthesis of secondary metabolites, biosynthesis of amino acids, histidine metabolism, and valine, leucine, and isoleucine biosynthesis. This indicates that *B. velezensis* continues to exhibit antimicrobial effects against *C. gloeosporioides* by enhancing the synthesis of secondary metabolites, amino acids, and specific metabolic pathways (e.g., histidine metabolism and branched-chain amino acid biosynthesis), effectively inhibiting the growth of the *C. gloeosporioides*. In the late-stage (D6_DvsD6_CK group), the PPI network contained 74 nodes and 160 edges. Core genes such as pdhC, fliF, and acoL were significantly enriched in pathways such as biosynthesis of secondary metabolites, biosynthesis of amino acids, flagellar assembly, microbial metabolism in diverse environments, arginine biosynthesis, and ABC transporters. The two pathways, biosynthesis of secondary metabolites and biosynthesis of amino acids, indicate that *B. velezensis* is still involved in synthesizing and releasing antimicrobial-related substances. Additionally, microbial metabolism in diverse environments suggests that *B. velezensis* might adjust its metabolic pathways to adapt to different environmental conditions while enhancing its antimicrobial efficacy during competition with *C. gloeosporioides*. Overall, *B. velezensis* adopts a multifaceted strategy to enhance its biological control capabilities across the three stages. By bolstering the synthesis of secondary metabolites, amino acids, improving motility, and nutrient utilization, it effectively inhibits the growth of *C. gloeosporioides*.

The qRT-PCR verification of 30 randomly selected genes demonstrated a 100% consistency rate with transcriptome sequencing results (Figure 13), confirming the reliability of the data for further investigation of *B. velezensis*’ key pathways and differentially expressed genes.

## 4. Discussion

This study aimed to investigate the potential of *B. velezensis* as a biocontrol agent against *C. gloeosporioides*, which causes anthracnose disease in various crops, including walnut [38,39,40]. The effects of *B. velezensis* on *C. gloeosporioides* were investigated through the process of confrontation culture on PDA medium. The results demonstrated that *B. velezensis* caused breakage, enlargement, and twisting of the mycelium of *C. gloeosporioides*, effectively inhibiting its growth. These findings were consistent with previous research on *Bacillus* species [41,42]. Furthermore, Huang et al. [43] showed that *B. velezensis* HYEB5-6 exerts significant antifungal activity against *C. gloeosporioides* by regulating proteases and glucanases, which inhibits the germination of *C. gloeosporioides* conidia, suppresses hyphal growth, and impairs conidiogenesis.

Moreover, we explored the preventive effects of *B. velezensis* fermentation broth, at varying concentrations and frequencies, against *C. gloeosporioides* on walnut leaves. The experiment revealed that the growth of *C. gloeosporioides* on walnut leaves and fruits could be effectively prevented by applying the fermentation broth diluted 50 to 100 times and sprayed once every two to three days. Contrary to initial expectations, a higher concentration of the broth resulted in a reduced preventive effect. This observation aligns with the concept of hormesis, wherein a substance exhibits a stimulating effect at low doses and an inhibitory effect at high doses [44,45]. When we applied this optimal concentration and frequency to the walnut fruit experiment, the results showed excellent protection of the walnut fruit, with *C. gloeosporioides* failing to infect the walnut fruit. This implies that effective control of *C. gloeosporioides*, and hence, protection of the host, can be achieved by managing the concentration and frequency properly. Therefore, we believe that applying this method to other agricultural and forestry crops should also yield favorable results against anthracnose disease caused by *C. gloeosporioides*.

During the initial response stage (D2_DvsD2_CK), *B. velezensis* demonstrated significant antibacterial effects against *C. gloeosporioides* in a confrontational culture on PDA medium. This resulted in the inhibition of *C. gloeosporioides*’ hyphal growth, restricting its overall development [46,47]. Moreover, when subjected to leaf infiltration, *C. gloeosporioides* showed a remarkable decrease in infectivity after interacting with *B. velezensis*. At the transcriptional level, *B. velezensis* exhibited the highest number of differentially expressed genes during this stage, indicating a strategic adjustment of its metabolic activity to enhance its overall antimicrobial capabilities. The most significantly enriched KEGG pathways and PPI core genes were associated with biosynthesis of secondary metabolites, biosynthesis of amino acids, ribosome, flagellar assembly, and carbon metabolism. At this stage, *B. velezensis* and *C. gloeosporioides* were observed to be relatively distant on the PDA medium. The significant enrichment of the flagellar assembly pathway suggested that flagella might play a crucial role in *B. velezensis*’ antibacterial activity, potentially being involved in its positioning, motility, and environmental sensing. *Bacillus* species typically achieve chemotaxis through flagellar movement, leading to the hypothesis that *B. velezensis* sensed the presence of *C. gloeosporioides* through its flagella. Previous studies have shown that *Bacillus* species possess strong mobility, enabling them to rapidly colonize plant surfaces, form extensive biofilms, and occupy essential spatial niches, thereby monopolizing resources [48,49]. Similarly, research by Bacon et al. [50] demonstrated that *Bacillus subtilis* efficiently establishes colonization and reproduces in maize, gaining an advantage in spatial niches, and effectively reducing the abundance of the pathogenic fungus *Fusarium moniliforme*. It is well known that many species of the *Bacillus* genus possess the ability to synthesize diverse secondary metabolites, including volatile organic compounds, non-ribosomal peptides, and antimicrobial proteins [51,52,53]. Additionally, considering the relatively distant proximity between *B. velezensis* and *C. gloeosporioides*, it is possible that *B. velezensis* exerted its antifungal effect through volatile organic compounds (VOCs) against *C. gloeosporioides* [54]. Yuan et al. [55] discovered that Bacillus amyloliquefaciens NJN-6 produces 36 VOCs, completely inhibiting the mycelial growth and conidial germination of *Fusarium oxysporum* f. sp. cubense, the causal agent of banana wilt disease, when treated with 200 µg of benzothiazole phenol, 2,3,6-trimethylphenol, 2-nonanone, or 2-undecanone. *B. velezensis*’ antifungal strategy is often multifaceted. The significant enrichment of the biosynthesis of secondary metabolites, biosynthesis of amino acids, and ribosome pathways in *B. velezensis* further supported its modulation of metabolic pathways to synthesize antimicrobial proteins and metabolites, thereby suppressing further invasion and growth of *C. gloeosporioides* [42].

During the intermediate stage (D4_DvsD4_CK), *B. velezensis* continued to exhibit significant antifungal effects against *C. gloeosporioides*, resulting in the inhibition of its normal growth and adopting a rhomboid-shaped morphology on PDA medium. Leaf inoculation experiments also indicated that *B. velezensis* significantly reduced the infectivity and toxicity of *C. gloeosporioides*. Transcriptome analysis revealed that during this stage, the most significantly enriched metabolic pathways based on KEGG functional annotation and PPI core gene enrichment were Biosynthesis of secondary metabolites, Biosynthesis of amino acids, Microbial metabolism in diverse environments, and ABC transporters. These findings suggest that *B. velezensis* further adjusted its metabolic pathways to enhance the production and transportation of metabolites, aiming to counteract *C. gloeosporioides* infection. As is known, *Bacillus* species can produce surfactants such as surfactin and iturin [56,57,58,59]. Interestingly, in the transcriptomic data, the srfAD genes crucial for surfactant synthesis in *B. velezensis* were found to be downregulated, indicating the capability of *B. velezensis* to synthesize surfactants and regulate this metabolic pathway in response to *C. gloeosporioides* infection [60,61,62,63]. In contrast, Caroline et al. [31] reported an up-regulation of the srfA gene during antimicrobial processes in *Bacillus* species. Despite the differential gene expression, both findings emphasize the ability of *B. velezensis* to precisely regulate the expression of srfAD and other related genes to synthesize surfactants and control this metabolic pathway in response to the infection caused by the pathogen. The significant enrichment of biosynthesis of secondary metabolites, biosynthesis of amino acids, and ABC transporters further corroborates the continuous antifungal effect of *B. velezensis* against *C. gloeosporioides* at the molecular level. Notably, at the same stage, genes involved in the initiation and regulation of spore formation, such as spore IIAA and spore IIAB, were found to be upregulated [64]. The production of spores is crucial for the survival of *Bacillus* under adverse conditions [65,66]. In addition to the significant enrichment of Microbial metabolism in diverse environments, this suggests *B. velezensis*’ adaptive changes and metabolic adjustments in response to the challenges posed by *C. gloeosporioides*. The activation of these genes implies that *B. velezensis* might form spores to enhance its resistance and adaptability against the anthracnose pathogen. These spores can remain dormant and viable for an extended period, enabling *B. velezensis* to withstand unfavorable conditions and survive longer than the pathogen, ensuring its survival and continued antimicrobial activity when conditions become favorable [67].

In the late phase (D6_DvsD6_CK group), *B. velezensis* continued to antagonize *C. gloeosporioides* on PDA medium, resulting in the continued rhomboid-shaped growth of *C. gloeosporioides*. Leaf infection assays indicated that after six days of confrontation, *C. gloeosporioides* exhibited the lowest infection ability at three time points. This suggests that after six days of continuous confrontation, a significant portion of *Colletotrichum gloeosporioides* hyphae were distorted, leading to a substantial reduction in its virulence. Transcriptome analysis revealed that the most significantly enriched metabolic pathways based on KEGG functional enrichment and PPI core gene enrichment were: microbial metabolism in diverse environments, carbon metabolism, biosynthesis of secondary metabolites, biosynthesis of amino acids, and flagellar assembly. Notably, the most significant enrichment shifted from biosynthesis of secondary metabolites to microbial metabolism in diverse environments and carbon metabolism. Moreover, transcriptome analysis revealed differential expression of the rbsK and rbsC genes, which play a significant role in enhancing *B. velezensis*’ efficient utilization of environmental nutrients [68]. This indicates a significant shift in the strategy of *B. velezensis* during this stage, with a focus on environmental adaptation, including the enhancement of carbon metabolism pathways for more efficient utilization of available carbon sources. Additionally, the enrichment of the flagellar assembly pathway, along with differential expression of core genes such as fliF and flgE, resulted in increased *B. velezensis* motility, which is crucial for flagellar assembly [69,70,71]. According to the theory proposed by Tohru [72], this increased motility enables *B. velezensis* to better perceive and respond to environmental cues, navigating towards favorable survival conditions. In summary, *B. velezensis* adjusted its gene expression to enhance motility, carbohydrate utilization efficiency, and energy production. These adaptations enable *B. velezensis* to improve its colonization ability, effectively compete with *Colletotrichum gloeosporioides*, and outcompete other microorganisms in the environment. This dynamic response equips *B. velezensis* to withstand the challenges posed by *C. gloeosporioides* and ensures its survival and success within the microbial community.

Our study bolstered evidence for *B. velezensis* as potent biocontrol agents against *C. gloeosporioides*, encouraging further agricultural research and applications. Future research was suggested to prioritize functional studies, such as gene knockout or overexpression experiments, to validate the identified key genes and pathways and unravel the complex regulatory networks involved in biocontrol. Additionally, researchers were recommended to explore comprehensive management strategies for tree diseases such as anthracnose by combining *B. velezensis* with other biological control agents or chemical reagents, aiming to achieve efficient control.

## 5. Conclusions

This study highlighted *B. velezensis*’ potential as an effective biocontrol agent against the growth and virulence of *C. gloeosporioides*. We showed that a well-tuned spray of *B. velezensis* fermentation broth could serve as a protective shield for walnut leaves and fruits, reducing the risk of walnut anthracnose spread. Through transcriptome analysis, we discovered that *B. velezensis* employs multiple synergistic mechanisms to comprehensively control plant pathogens. For instance, it enhances the synthesis of metabolic products, improves flagellar assembly and chemotaxis, and enhances nutrient utilization capabilities, leading to a successful inhibition of *C. gloeosporioides*. Further investigations are required to precisely identify the biocontrol role of these crucial genes.

## Figures and Tables

**Figure 1 microorganisms-11-01885-f001:**
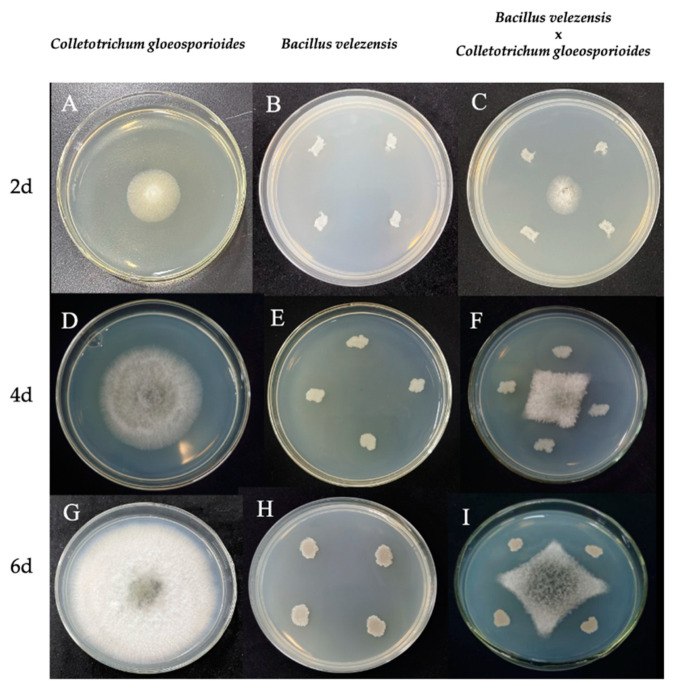
The mycelial morphology of *C. gloeosporioides* and *B. velezensis* on PDA medium after 2, 4, and 6 days of culture. (**A**,**D**,**G**) *C. gloeosporioides* cultured alone. (**B**,**E**,**H**) *B. velezensis* cultured alone. (**C**,**F**,**I**) Confrontation culture of *C. gloeosporioides* and *B. velezensis* strain. (**A**–**C**) Cultured on the second day. (**D**–**F**) Cultured on the fourth day. (**G**–**I**) Cultured on the sixth day.

**Figure 2 microorganisms-11-01885-f002:**
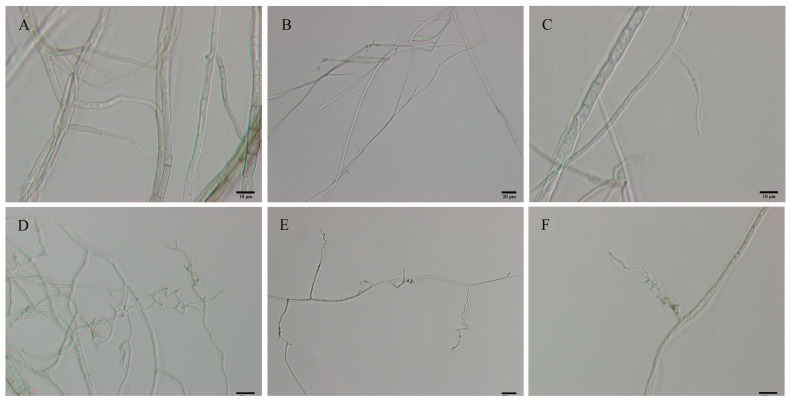
Morphological characteristics of *C. gloeosporioides* mycelium. (**A**–**C**) *C. gloeosporioides* mycelial morphology on PDA medium alone, 6 days after inoculation; (**D**–**F**) *C. gloeosporioides* mycelial morphology co-cultured with *B. velezensis* on PDA medium, 6 days after inoculation. Scale bars: (**A**,**C**,**D**,**F**) = 10 μm; (**B**,**E**) = 20 μm.

**Figure 3 microorganisms-11-01885-f003:**
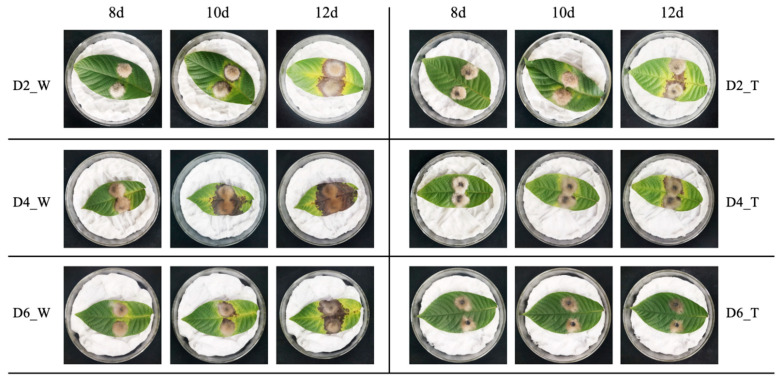
Changes in the infectivity of *C. gloeosporioides* on walnut leaves after co-cultivation with *B. velezensis* for different durations. D2_W, D4_W, and D6_W represented the infection of *C. gloeosporioides* on walnut leaves cultured alone for 2, 4, and 6 days, respectively. D2_T, D4_T, and D6_T represent the infection of walnut leaves by *C. gloeosporioides* after co-culturing with *B. velezensis* for 2, 4, and 6 days, respectively. 8d, 10d and 12d represent the infection results of leaves on the 8th, 10th, and 12th day, respectively. Each treatment and control had three replicates.

**Figure 4 microorganisms-11-01885-f004:**
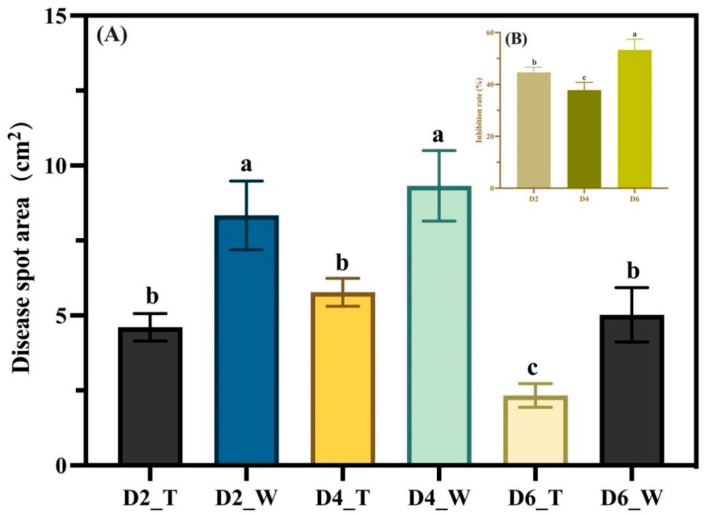
Lesion area and inhibition rate. (**A**) Lesion areas of leaves in three control and three treatment groups measured on the 12th day after infection; (**B**) Inhibition rate results calculated from the lesion areas measured on the 12th day. Data were analyzed using a one-way ANOVA test for comparison between groups, revealing a statistically significant difference between the groups (*p* < 0.05). Bars above columns represent standard errors. Different letters indicate significant differences between the treatments (*p* < 0.05).

**Figure 5 microorganisms-11-01885-f005:**
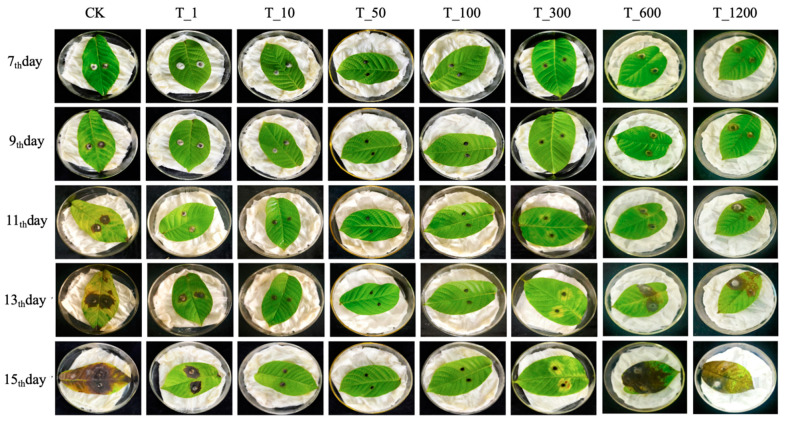
Changes of disease spots in walnut leaves from the 7th to the 15th day after control with different concentrations of fermentation broth. Note: Seven concentration gradients of *B. velezensis* spray were used: 1, 10, 50, 100, 300, 600, and 1200 times. The treatment groups were labeled T_1, T_10, T_50, T_100, T_300, T_600, and T_1200, with CK as the control. The vertical sequence represents the co-cultivation time: 7th, 9th, 11th, 13th, and 15th day. Each treatment and control had three replicates.

**Figure 6 microorganisms-11-01885-f006:**
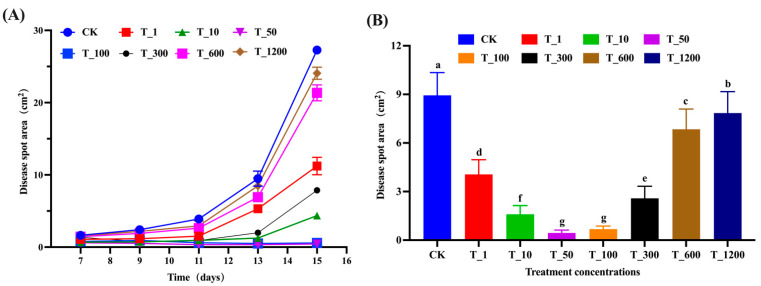
The effect of spraying *B. velezensis* fermentation broth at different concentrations on the size of disease spots on walnut leaves. (**A**) The lesion area of walnut leaves in different treatment groups at different time points. The data were from five sampling times and eight treatments. The values on the figure were the average of three replicates; (**B**) The data are the mean values of different concentration treatment groups at five time points, different letters in panel (**B**) denote significant differences between treatments based on repeated-measures ANOVA (*p* < 0.05). Seven concentration gradients of *B. velezensis* spray were used: 1, 10, 50, 100, 300, 600, and 1200 times. The treatment groups were labeled T_1, T_10, T_50, T_100, T_300, T_600, and T_1200, with CK as the control. The vertical sequence represents the co-cultivation time: 7th, 9th, 11th, 13th, and 15th day. Each treatment and control had three replicates.

**Figure 7 microorganisms-11-01885-f007:**
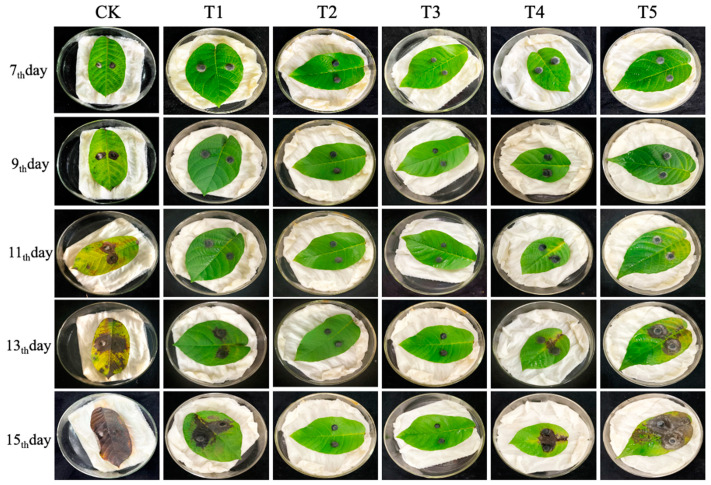
Impact of treatment frequency on the lesion area of walnut leaves. Note: *B. velezensis* spray frequency was set at five gradients: T1, T2, T3, T4, and T5, representing spraying every 1 day, 2 days, 3 days, 4 days, and 5 days, respectively. The vertical sequence represents the co-cultivation time: 7th, 9th, 11th, 13th, and 15th day. Each treatment and control had three replicates.

**Figure 8 microorganisms-11-01885-f008:**
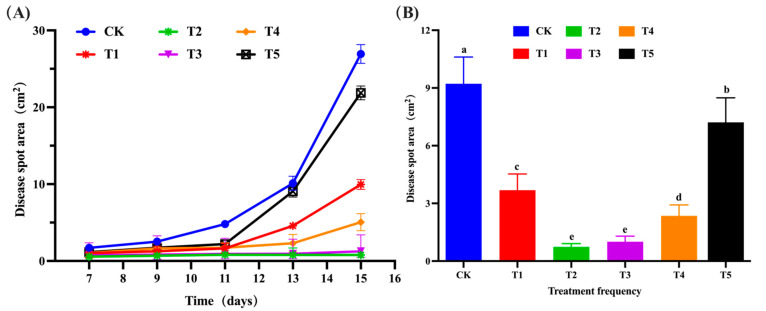
The effect of spraying *B. velezensis* fermentation broth at different frequencies on the size of disease spots on walnut leaves. (**A**) Lesion area of walnut leaves in different treatment groups at various time points and frequencies. Data were obtained from five samplings and six treatments. The values in the figure represent the average of three replicates; (**B**) The data display the mean values of different concentration treatment groups at 5 time points. In Panel (**B**), different letters indicate significant differences between treatments based on repeated measures analysis of variance (*p* < 0.05).

**Figure 9 microorganisms-11-01885-f009:**
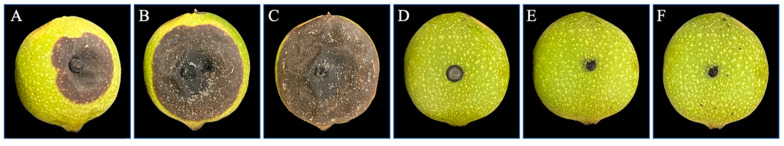
Control effect of *B. velezensis* on *C. gloeosporioides* in walnut fruit. (**A**–**C**) the control group; (**D**–**F**) the treatment group; (**A**,**D**) photographed on the 14th day; (**B**,**E**) photographed on the 16th day; (**C**,**F**) photographed on the 18th day.

**Figure 10 microorganisms-11-01885-f010:**
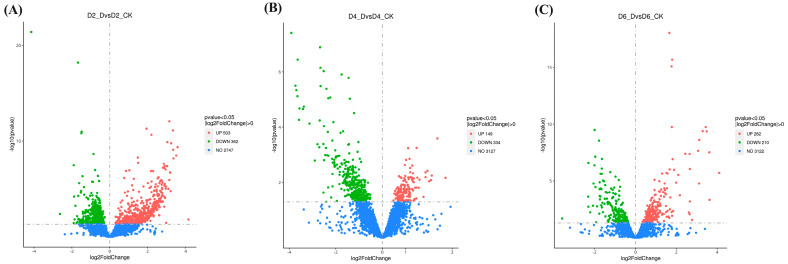
Volcano plot of DEGs in compared groups. (**A**) D2_DvsD2_CK group; (**B**) D4_DvsD4_CK group; (**C**) D6_DvsD6_CK group. Note: D2_DvsD2_CK, D4_DvsD4_CK, and D6_DvsD6_CK groups represent the comparisons between *B. velezensis* cultured alone for 2, 4, and 6 days, respectively, and *B. velezensis* co-cultured (confrontation with *C. gloeosporioides*) for the same duration. DEGs were selected based on the criteria of *p* < 0.05 and |log2FoldChange| > 0. The *X*-axis (log2FoldChange) indicates the fold change in gene expression, and the *Y*-axis (−log10(*p*-value)) represents the significance level of gene expression differences among the groups. Red dots indicate upregulated genes, green dots indicate downregulated genes, and blue dots represent genes with no significant expression changes.

**Figure 11 microorganisms-11-01885-f011:**
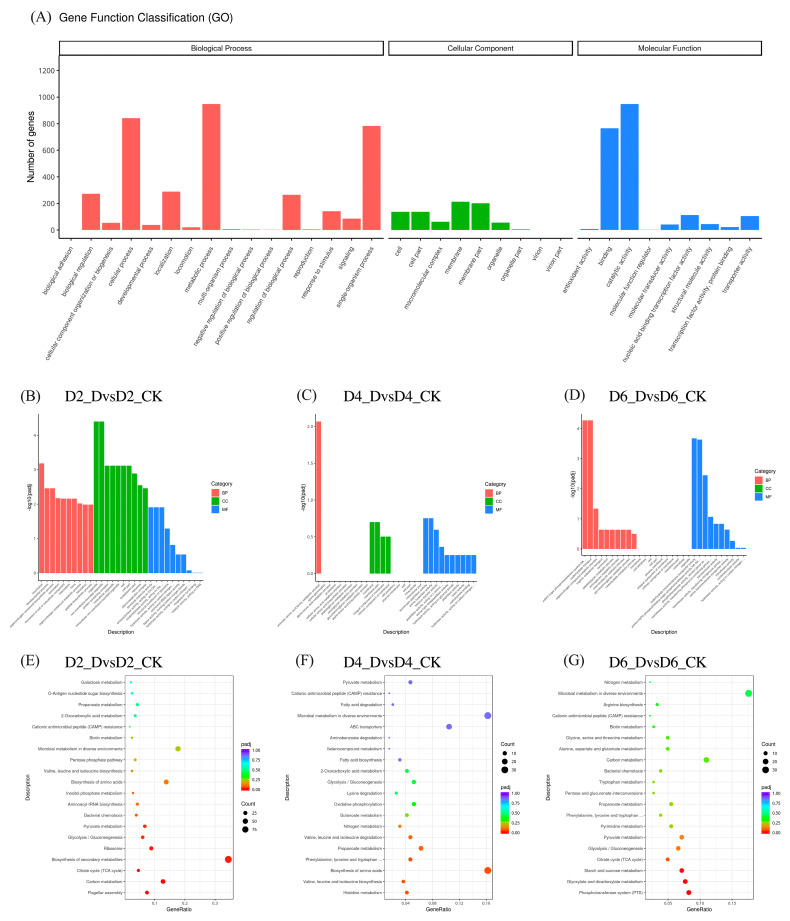
Histogram of GO functional enrichment analysis and bubble chart of KEGG pathway enrichment analysis of DEGs in compared groups. Note: (**A**–**D**) Histogram of GO functional enrichment analysis of DEGs in compared groups; (**A**) Functional classification of overall DEGs; (**B**) D2_DvsD2_CK group; (**C**) D4_DvsD4_CK group; (**D**) D6_DvsD6_CK group. In graphs (**A**–**D**), the *x*-axis represents GO Terms, and the *y*-axis represents the significance level of GO Term enrichment, represented as −log10(padj). Different colors denote distinct functional categories: red for biological processes, green for cellular components, and blue for molecular functions. (**E**–**G**) Bubble chart of KEGG pathway enrichment analysis of DEGs in compared groups; (**E**) D2_DvsD2_CK group; (**F**) D4_DvsD4_CK group; (**G**) D6_DvsD6_CK group. The bubble chart shows the top 20 significantly enriched KEGG pathways. The *x*-axis represents the ratio of differentially expressed genes annotated to each pathway to the total differentially expressed genes. The *y*-axis represents the KEGG pathways, with bubble size indicating the number of annotated genes. Larger circles denote more enriched differentially expressed genes. The color gradient (red to purple) represents the enrichment significance.

**Figure 12 microorganisms-11-01885-f012:**
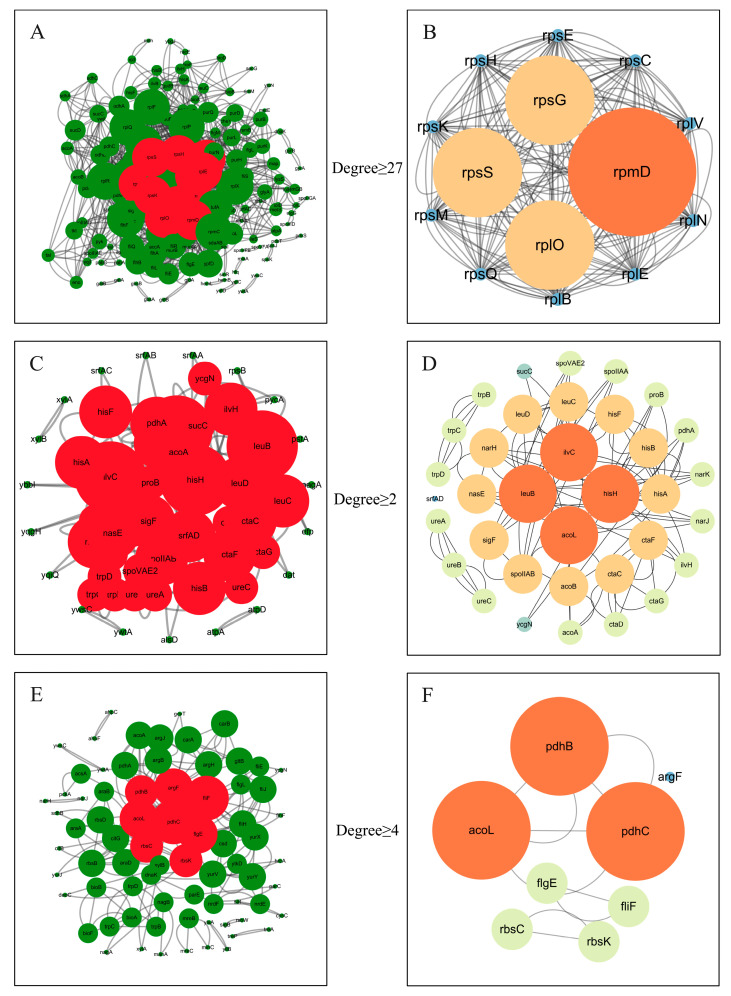
PPI network diagram results. (**A**,**B**) D2 Dvs D2 CK group; (**C**,**D**) D4 DvsD4 CK group; (**E**,**F**) D6 DvsD6 CK group.

**Figure 13 microorganisms-11-01885-f013:**
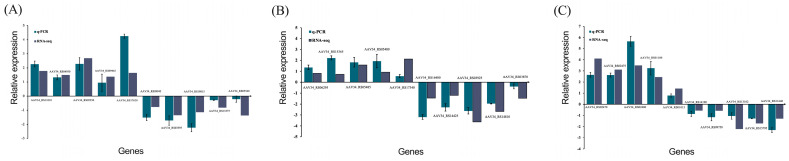
Histogram of relative expression of DEGs by qRT-PCR and transcriptome analysis. (**A**) D2 DvsD2 CK group; (**B**) D2_DvsD2 CK group; (**C**) D2 DvsD2 CK group.

## Data Availability

The data from this study are openly available in the NCBI’s Sequence Read Archive (SRA) database under the accession number PRJNA742867. You can access the data at https://www.ncbi.nlm.nih.gov/bioproject/?term=PRJNA742867 (accessed on 1 July 2021).

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
