# Peer review of "Strong Opponent of Walnut Anthracnose—Bacillus velezensis and Its Transcriptome Analysis"

_microorganisms, 2023, doi:10.3390/microorganisms11081885_

Round 1

Reviewer 1 Report

This is an interesting manuscript on the characterisation of Bacillus velezensis as a biocontrol agent for the pathogen Colletotrichum gloeosporioides, which causes walnut anthracnose. The authors performed a transcriptional analysis of the bacterium during the interaction. The study is well-designed, the procedures are correct, and the results are of good quality. I believe that the work is suitable for publication in Microorganisms. However, I have the following comments to improve the scientific soundness of the manuscript.

- A general problem with the figures in this paper is the very poor information in the legends. The figures must stand out from the text and clearly convey the main findings. Please provide a brief description of the materials and methods used and a brief explanation of the results for each figure.

- Revise the font size and the units for quantities.

- Clarify the meaning of the different colours and sizes of the symbols as in bubble charts.

- The scale bar in Figure 2 needs to be more visible; text resolution is a big problem in Figure 11.

- The labelling of the Y-axis in Figure 13 is missing.

For the qPCR experiment, the authors must explain why they chose GAPDH as the reference gene. Has GAPDH been validated as a reliable housekeeping gene in Bacillus velezensis? Is there a reference? Furthermore, the calculation of the efficiency of the qPCR primers is not given.

We found minor grammatical errors that do not seriously affect the understanding of the manuscript. However, we suggest that the authors get editing help from someone with full professional proficiency in the English language.

Author Response

Response to Reviewer 1 Comments

Dear reviewer,

Thank you for your invaluable comments and suggestions. They have greatly contributed to refining and improving our paper, providing significant guidance for our research. We sincerely apologize for any inconvenience caused by our inexperience in the submission process. We are fully committed to humbly accepting your feedback and diligently revising our manuscript until it meets your satisfaction. We have thoroughly reviewed your comments and made the corrections accordingly. We hope that these changes address all the concerns raised and meet with your approval. Should there be any further areas requiring improvement, please do not hesitate to point them out, and we will diligently address them until you are completely satisfied. Once again, we sincerely appreciate your understanding and assistance throughout this process. Below, we provide a summary of the main corrections made in the paper and our responses to the reviewer's comments:

Q1. A general problem with the figures in this paper is the very poor information in the legends. The figures must stand out from the text and clearly convey the main findings. Please provide a brief description of the materials and methods used and a brief explanation of the results for each figure.

Response: Thank you for your valuable suggestions; we greatly appreciate them! Following your advice, we have provided detailed descriptions and made necessary modifications to the figures and corresponding sections of the main text. We hope that these revisions meet your expectations. If there are any remaining areas of concern, please feel free to point them out, and we will continue to make further improvements until you are completely satisfied.

Q2. Revise the font size and the units for quantities.

Response: We sincerely apologize for not thoroughly checking these details earlier and any inconvenience it may have caused you. This time, we have carefully reviewed and addressed the font and quantity units, making the necessary modifications. Thank you for bringing this to our attention and for your valuable reminders. We greatly appreciate your understanding and assistance.

Q3. Clarify the meaning of the different colours and sizes of the symbols as in bubble charts.

Response: Thank you for your valuable suggestions, which we wholeheartedly embrace. We have added detailed explanations below Figure 11, clarifying the meanings of the X and Y coordinates, different colors, and symbol sizes, aiming to provide readers with a clear understanding. We hope this meets your expectations, and if there are any further areas that need improvement, we will continue to make necessary modifications. Thank you once again for your kind guidance!

Q4. The scale bar in Figure 2 needs to be more visible; text resolution is a big problem in Figure 11.

Response: We sincerely apologize for any inconvenience caused earlier. We have made modifications to Figure 2, highlighting the scale bar and providing annotations below the image to indicate the corresponding scale size. Additionally, we have added detailed explanations to Figure 11, describing the specific meanings of each subfigure, as well as the representations of colors and shapes. If there are any shortcomings, please do not hesitate to point them out, and we will continue to make adjustments until meeting the requirements. Thank you for your valuable guidance!

Q5. The labelling of the Y-axis in Figure 13 is missing.

Response: Thank you for your reminder. We have re-added the labels for the Y-axis, and we sincerely apologize for the oversight. If there are any other omissions or areas that require improvement, please let us know, and we will promptly address them. Your valuable feedback is greatly appreciated!

Q6. For the qPCR experiment, the authors must explain why they chose GAPDH as the reference gene. Has GAPDH been validated as a reliable housekeeping gene in Bacillus velezensis? Is there a reference? Furthermore, the calculation of the efficiency of the qPCR primers is not given.

Response: Thank you for your valuable suggestions. Following your advice, we have included the rationale for choosing the GAPDH gene in section 2.4.4 of the manuscript and cited the relevant reference. The chosen reference is a study we consulted before conducting our experiments, where the author used GAPDH as a reference gene for Bacillus amyloliquefaciens. Given the close phylogenetic relationship between Bacillus amyloliquefaciens and Bacillus velezensis, and the well-known stability of GAPDH as a housekeeping gene in various organisms, including fungi, bacteria, and plants, we decided to explore its potential as a reference gene in our q-PCR experiments.

Prior to conducting the q-PCR experiments, we performed a pilot study to assess the expression of the GAPDH gene in Bacillus velezensis. The results of the pilot study demonstrated that the q-PCR results for the GAPDH gene in different treatment groups were highly stable, showing minimal variation. This finding confirms that GAPDH is a reliable and stable housekeeping gene in our experimental conditions. As a result, we have chosen GAPDH as the reference gene for our subsequent q-PCR experiments.

We hope that these additional details address your concerns and provide a comprehensive explanation for the selection of GAPDH as the reference gene in our study. Thank you again for your valuable feedback.

Thank you again for your time and consideration. We look forward to hear from you soon.

With kind regards

Linmin Wang & Tianhui Zhu

Reviewer 2 Report

The work I reviewed, „Strong opponent of walnut anthracnose -- Bacillus velezensis and its transcriptome analysis“ is an extensive and detailed study of the Bacillus velezensis strain use as a biocontrol agent. From the title and the experiments performed, it is clear that the authors had the idea to delve a little deeper into the molecular mechanisms and explain what happens at this level when these two strains (Bacillus velezensis and Colletotrichum gloeosporioides, anthracnose pathogen. the most widespread and serious postharvest disease of many fruits including mango, papaya, pitaya, and avocado) come into contact with each other. The entire study was excellently designed and implemented. A large number of experiments were performed with a sufficient number of replicates and controls. In addition, a large number of results were obtained, so we are witnessing a large number of Figures included in the work and numerous calls for supplementary material. I must dissociate myself from this and say that I had no insight into the supplementary material, that it was not available in the work I reviewed, and that some additional documents available could not be opened.

General comments

The text itself contains some technical, orthographic, or stylistic errors that need to be corrected before publication. E.g.:

Line 94 – there are some unnecessary spaces;

Line 159 – It is written: „In method 4.2....“ and I think it should be „In method 2.2...“

The text generally uses many abbreviations and introduced symbols for experimental settings, which makes it difficult to follow the text. It is necessary to remind readers from time to time in the text what each designation stands for, i.e., what part of the text and what settings it refers to.

The captions under Figures 3, 5, 6, and 10 should be more detailed and explain the markings in the figure. It is important that the figure can be followed and understood without having to go back to the text and look up what each mark means.

Minor revision

There is now Figure 8 in the text. Please correct that. Please provide additional information on the biocontrol strain Bacillus velezensis, where it was isolated, whether or not it has already been characterized for some biocontrol characteristics, etc.

I think the discussion needs to be improved. The transcriptomics results are the most discussed, but they are not related to the experimental results. The strain is an extremely potent biocontrol agent, but this was not explained by the transcriptomics analysis. One section comments on the results for surfactin genes and that's it. The spectrum of synthesized lipopeptides is large, so it is not sufficient to look for an explanation of the activity of a single lipopeptide.

It is also generally better to explain in more detail the extremely good biocontrol potential of the analyzed strain. In this way, the whole work seems to be unfinished.

Author Response

Response to Reviewer 2 Comments

Dear reviewer,

Thank you for your invaluable comments and suggestions. They have greatly contributed to refining and improving our paper, providing significant guidance for our research. We sincerely apologize for any inconvenience caused by our inexperience in the submission process. We are fully committed to humbly accepting your feedback and diligently revising our manuscript until it meets your satisfaction. We have thoroughly reviewed your comments and made the corrections accordingly. We hope that these changes address all the concerns raised and meet with your approval. Should there be any further areas requiring improvement, please do not hesitate to point them out, and we will diligently address them until you are completely satisfied. Once again, we sincerely appreciate your understanding and assistance throughout this process. Below, we provide a summary of the main corrections made in the paper and our responses to the reviewer's comments:

Q1. A large number of experiments were performed with a sufficient number of replicates and controls. In addition, a large number of results were obtained, so we are witnessing a large number of Figures included in the work and numerous calls for supplementary material. I must dissociate myself from this and say that I had no insight into the supplementary material, that it was not available in the work I reviewed, and that some additional documents available could not be opened.

Response: We sincerely apologize for the inconvenience caused by not providing the supplementary materials earlier. We deeply regret any inconvenience this may have caused you. This time, we will upload the additional materials for your review. Additionally, if you encounter any files that you are unable to open due to formatting issues, please inform us immediately, and we will promptly adjust the formats to offer you different options. Our aim is to ensure you have a pleasant reviewing experience. Your valuable feedback has been instrumental in continuously improving and bringing our article closer to publication standards. Thank you once again for all your feedback. It has contributed significantly to the refinement of our work. Much appreciated!

Q2. Line 159 – It is written: „In method 4.2....“ and I think it should be „In method 2.2...

Response: We sincerely apologize for the oversight, and thank you for bringing it to our attention. We have now made the necessary corrections, and the correct phrase should be "In method 2.2." Please rest assured that we will be more attentive to such details in the future. Your understanding and patience are greatly appreciated. If you have any further suggestions or concerns, please feel free to let us know. Thank you for your valuable feedback!

Q3. The text generally uses many abbreviations and introduced symbols for experimental settings, which makes it difficult to follow the text. It is necessary to remind readers from time to time in the text what each designation stands for, i.e., what part of the text and what settings it refers to.

Response: Thank you for your valuable feedback; it has been instrumental in significantly improving our paper. We have followed your advice and made the necessary revisions, including adding detailed explanations below each figure and reiterating the treatment group identifiers in the Results section. This will help readers quickly grasp the meaning of each symbol used in the figures without having to refer back to the Methods section, streamlining the reading experience and enhancing the clarity of the manuscript. We hope these changes meet your expectations.

Your guidance and support have been greatly appreciated. If you have any further suggestions or concerns, please do not hesitate to let us know. Thank you for your valuable input!

Q4. The captions under Figures 3, 5, 6, and 10 should be more detailed and explain the markings in the figure. It is important that the figure can be followed and understood without having to go back to the text and look up what each mark means.

Response: Thank you very much for your valuable suggestions. We have followed your advice and made detailed revisions, providing comprehensive annotations below Figures 2, 3, 5, 6, 7, 10, and 11. This will enable readers to quickly understand the corresponding treatment groups, the specific meanings of the coordinate axes, and the implications of symbol sizes and colors used in the figures. We hope these enhancements meet your expectations. If there are any shortcomings, please kindly point them out, and we will promptly make the necessary adjustments until meeting the requirements. Your feedback is highly appreciated, and we are committed to improving the manuscript to the best of our ability. Thank you once again for your valuable input!

Q5. There is now Figure 8 in the text. Please correct that.

Response: We sincerely apologize for the oversight, and thank you for bringing it to our attention. We have now made the necessary corrections to the format of Figure 8, as per your requirements. Please review it, and we hope it now meets your expectations. Your understanding and patience are greatly appreciated. If you have any further suggestions or concerns, please feel free to let us know. Thank you for your valuable feedback!

Q6. Please provide additional information on the biocontrol strain Bacillus velezensis, where it was isolated, whether or not it has already been characterized for some biocontrol characteristics, etc.

Response: Thank you for your reminder. We have added the source of Bacillus velezensis strain, as per your request. After isolating and identifying Bacillus velezensis, we conducted several experiments, including using 7 primers for identification, constructing a phylogenetic tree, and conducting physiological and biochemical characterization. Additionally, through further antagonistic experiments, we found that Bacillus velezensis exhibits varying degrees of inhibitory activity against four pathogenic fungi: Fusarium solani, Phomopsis capsici, Neofusicoccum parvum, and Botryosphaeria dothidea. However, due to space limitations in this paper, we were unable to include all of this data. We appreciate your understanding. If you have any further suggestions or guidance, we will promptly consider and implement them. Thank you for your valuable input!

Q7. I think the discussion needs to be improved. The transcriptomics results are the most discussed, but they are not related to the experimental results. The strain is an extremely potent biocontrol agent, but this was not explained by the transcriptomics analysis. One section comments on the results for surfactin genes and that's it. The spectrum of synthesized lipopeptides is large, so it is not sufficient to look for an explanation of the activity of a single lipopeptide.

Response: First of all, we sincerely apologize for any inconvenience caused. We have carefully reviewed and reanalyzed the transcript section as per your feedback. If you feel that it still needs further improvement, please don't hesitate to point it out. We are more than willing to continue making revisions until you are completely satisfied. Thank you again for your valuable input. Much appreciated!

Thank you again for your time and consideration. We look forward to hear from you soon.

With kind regards

Linmin Wang & Tianhui Zhu
